# Screening and Relative Quantification of Migration from Novel Thermoplastic Starch and PBAT Blend Packaging

**DOI:** 10.3390/foods14132171

**Published:** 2025-06-21

**Authors:** Phanwipa Wongphan, Elena Canellas, Cristina Nerín, Carlos Estremera, Nathdanai Harnkarnsujarit, Paula Vera

**Affiliations:** 1Department of Packaging and Materials Technology, Faculty of Agro-Industry, Kasetsart University, 50 Ngam Wong Wan Rd., Latyao, Chatuchak, Bangkok 10900, Thailand; phanwipa.w@ku.th; 2GUIA Group, Department of Analytical Chemistry, University of Zaragoza, I3A, María de Luna, 3, 50018 Zaragoza, Spain; ecanellas@unizar.es (E.C.); cnerin@unizar.es (C.N.); cjimenez@unizar.es (C.E.); 3Center for Advanced Studies for Agriculture and Food, Kasetsart University, 50 Ngam Wong Wan Rd., Latyao, Chatuchak, Bangkok 10900, Thailand

**Keywords:** migration, biodegradable, non-intentionally added substances, ultra-high-pressure liquid chromatography coupled with quadrupole time-of-flight mass spectrometry, food packaging

## Abstract

A novel biodegradable food packaging material based on cassava thermoplastic starch (TPS) and polybutylene adipate terephthalate (PBAT) blends containing food preservatives was successfully developed using blown-film extrusion. This active packaging is designed to enhance the appearance, taste, and color of food products, while delaying quality deterioration. However, the incorporation of food preservatives directly influences consumer perception, as well as health and safety concerns. Therefore, this research aims to assess the risks associated with both intentionally added substances (IAS) and non-intentionally added substances (NIAS) present in the developed active packaging. The migration of both intentionally and non-intentionally added substances (IAS and NIAS) was evaluated using gas chromatography–mass spectrometry (GC-MS) and ultra-high-performance liquid chromatography coupled with quadrupole time-of-flight mass spectrometry (UHPLC-Q-TOF-MS). Fifteen different volatile compounds were detected, with the primary compound identified as 1,6-dioxacyclododecane-7,12-dione, originating from the PBAT component. This compound, along with others, resulted from the polymerization of adipic acid, terephthalic acid, and butanediol, forming linear and cyclic PBAT oligomers. Migration experiments were conducted using three food simulants—95% ethanol, 10% ethanol, and 3% acetic acid—over a period of 10 days at 60 °C. No migration above the detection limits of the analytical methods was observed for 3% acetic acid and 10% ethanol. However, migration studies with 95% ethanol revealed the presence of new compounds formed through interactions between the simulant and PBAT monomers or oligomers, indicating the packaging’s sensitivity to high-polarity food simulants. Nevertheless, the levels of these migrated compounds remained below the regulatory migration limits.

## 1. Introduction

The European Union has recently emphasized the significant potential of biobased and biodegradable products to foster more sustainable economies and reduce dependence on fossil fuels, presenting promising growth opportunities for the industry. Biomaterials, particularly bioplastics, are garnering increasing attention in the food industry for packaging applications. European Bioplastics defines bioplastics as materials either sourced from renewable or biological resources or as biodegradable plastics [1,2]. Additionally, the packaging industry is actively seeking more eco-friendly alternatives to conventional materials, driving researchers to focus on developing biodegradable and bioplastic-based food packaging. Common materials used in this area include thermoplastic starch (TPS), polybutylene adipate terephthalate (PBAT), polylactide (PLA), and polymer blends [3,4,5].

This work focuses on biodegradable films made from thermoplastic starch (TPS) blended with polybutylene adipate terephthalate (PBAT). PBAT is a random copolyester synthesized via a polycondensation reaction involving adipic acid, 1,4-butanediol, and dimethyl terephthalate. Although derived from petroleum-based raw materials, PBAT is biodegradable. It consists of two structural segments: (i) polybutylene adipate (PBA), an aliphatic component, and (ii) polybutylene terephthalate (PBT), an aromatic component, thereby incorporating both aliphatic and aromatic characteristics [6,7]. It is regarded as an excellent and highly inert material with excellent flexibility, thermal stability, and biodegradability, which are comparable to those of low-density polyethylene (LDPE) [8,9,10]. However, PBAT had limitations such as a high cost and slow and longer degradation cycle than other biodegradable plastics because of the butylene group. For TPS, as starch-based materials are highly hydrophilic, this leads to poor mechanical strength, formability, and stability [3,7,11]. Therefore, polyester/starch-based blends are among the most widely used biodegradable plastics for food packaging applications, particularly as food contact materials (FCMs) [4,5,12]. The development of new food packaging must prioritize safety by ensuring that no harmful compounds migrate from the packaging into the food [6,13,14].

The safety assessment and migration of substances from packaging needs to involve both intentionally added substances (IAS), such as plasticizers, slip agents, monomers, additives, and solvents, as well as non-intentionally added substances (NIAS), which include degradation products formed during or after manufacturing, impurities in raw materials, and interaction products between packaging components [15,16]. Therefore, the aim of this article is to assess the risk associated with IAS and NIAS present in the developed active packaging from our previous work [3]. Migration screening was conducted using gas chromatography coupled with mass spectrometry (GC-MS) and ultraperformance liquid chromatography quadrupole-time-of-fight mass spectrometry (UPLC-Q-TOF–MS) to analyze the migration from biodegradable TPS/PBAT samples intended for food packaging applications.

## 2. Materials and Methods

### 2.1. Chemicals and Reagents

1,4,7-Trioxacyclotridecane-8,13-dione (98% purity, CAS: 6607-37-7), 2,2,4-trimethyl-1,3-pentanediol diisobutyrate (≥98.5%, CAS: 6846-50-0), and n-alkane standards (C7–C40) were purchased from Sigma–Aldrich (Madrid, Spain). Absolute ethanol (HPLC grade, CAS: 64-17-5), acetic acid, and dichloromethane were supplied by Scharlau (Setmenat, Spain). Hexane was obtained from Fisher Chemicals (Gothenburg, Sweden Loughborough, UK). Ultrapure water was produced using a Wasserlab Ultramatic GR system (Barbatáin, Spain). Solid-phase microextraction (SPME) fibers, including polydimethylsiloxane (100 μm), polyacrylate (85 μm), and divinylbenzene/Carboxen/polydimethylsiloxane (50/30 μm), were supplied by Supelco (Madrid, Spain).

### 2.2. Sample Characteristics

Neat PBAT film and PBAT/TPS blended film incorporated additives, namely sodium nitrite, sodium erythorbate, tetrasodium pyrophosphate, sodium tripolyphosphate, and sodium hexametaphosphate, which were produced via blown-film extrusion. The names and characteristics of the samples are shown in Table 1. The averaged thickness of all film was 0.04 mm.

### 2.3. Sample Extraction

All films were cut into small pieces, and 0.5 g of each sample was extracted three times with 3 mL of solvent in an ultrasonic bath (frequency applied: 40 Hz, Branson Ultrasonics^TM^ 3510, Emerson Electric Co., St. Louis, MO, USA) at 40 °C for 1 h to enhance solvent penetration and disrupt the matrix, improving compound release. Dichloromethane and ethanol were used as extraction solvents to dissolve non-polar and polar volatile compounds, respectively. The extraction solution (total volume of 9 mL) was concentrated to 2 mL under nitrogen flow using a nitrogen evaporator (TECHNE sample concentrator, Cole-Parmer Ltd., Eaton Socon, UK) at 60–65 °C before injection. Each sample and solvent blank were analyzed in duplicate with blank extractions performed every two sample extractions to monitor contamination. In addition, PBAT resin and PBAT film were used as positive controls by controlling the weight of resin and film in the extraction to control the concentration of the extracted amount. Samples were analyzed using GC-MS with liquid injection.

### 2.4. Migration Assays

Migration analysis was carried out by total immersion of the film samples in three different food simulants: 10% ethanol (simulant A), 3% acetic acid (simulant B), and 95% ethanol (simulant D2), following the specifications of Regulation (EU) No. 10/2011. The films were cut into strips measuring 1 × 4 cm^2^ for simulant D2 and 1 × 5 cm^2^ for simulants A and B, ensuring the appropriate surface area-to-volume ratio. Each sample was placed in 20 mL glass vials and 18 mL of the corresponding simulant was added under gravimetric control. All samples were incubated in the oven at 60 °C for 10 days. Blank samples were prepared and treated under identical conditions. Migration tests were conducted in triplicate, with blank extractions performed after every three test samples. The simulant samples were analyzed using and UPLC-Q-TOF–MS to determine the migrated compounds from film samples. This offered a high-resolution mass analysis suitable for non-targeted and semi-targeted screening of migrants (e.g., additives, degradation products, and monomers).

### 2.5. Instrument Conditions

#### 2.5.1. GC-MS

Samples subjected to migration testing via liquid injection were analyzed using a 6890N gas chromatograph with a 5977D mass spectrometry detector (Agilent Technilogies, Santa Clara, CA, USA). Separation was performed on an HP-5MS capillary column (30 m × 0.25 mm i.d. × 0.25 μm film thickness) obtained from Agilent Technologies (Madrid, Spain).

A total of 1 μL of migration extract was injected in the splitless mode. High-purity helium (99.99%) was used as the carrier gas at a constant flow rate of 1.0 mL/min. The GC oven temperature program was as follows: an initial temperature of 50 °C was held for 5 min, followed by an increase to 300 °C at a ramp rate of 10 °C/min. A solvent delay of 5 min was applied. Data acquisition was performed in the SCAN mode. Volatile compounds were identified by comparing the acquired mass spectra with those in the National Institute of Standards and Technology (NIST) Chemistry WebBook spectral library integrated into the instrument software. The minimum requirement of mass spectrum match to consider a volatile compound by NIST was 80%. The standard of the compounds was analyzed for confirmation and quantitation using calibration curves from an external standard compound in ethanol, based on the same time period and having a chemical structure similar or close to that of the obtained substance.

#### 2.5.2. SPME-GC-MS

Samples were extracted using a divinylbenzene/Carboxen/polydimethylsiloxane (DVB/CAR/PDMS) solid-phase microextraction (SPME) fiber (50/30 μm). DVB/CAR/PDMS fiber (50/30 μm) is well-suited for a broad range of volatile and semi-volatile compounds due to its mixed sorbent phases, combining polarity, porosity, and surface area. Preliminary extractions were performed at 80 °C for 30 min to enhance analyte partitioning into the headspace without compromising thermal stability. The SPME analysis was conducted in a 20 mL glass vial, fully filled with simulants, under the same conditions used for GC-MS analysis.

#### 2.5.3. UHPLC-Q-TOF-MS Analysis

The samples were analyzed using Acquity^TM^ UHPLC chromatography system coupled to a Xevo G2 QTOF mass spectrometer (Waters, Manchester, UK). The mass spectrometer was equipped with an atmospheric pressure ionization (API) source operating with an electrospray ionization (ESI) interface. The detector configuration includes a hexapole, a quadrupole, a collision cell, and a time of flight as analyzer (QTOF). Chromatographic separation was performed using an ACQUITY UPLC^®^ BEH C18 column (2.1 × 100 mm, 1.7 μm particle size, Agilent Technologies Spain, S.L., Madrid, Spain) at a flow rate of 0.3 mL/min. The gradient elution was performed using ultrapure water with 0.1% formic acid as mobile phase A and methanol with 0.1% formic acid as mobile phase B. The gradient started at 5% phase B and increased linearly to 100% phase B over 15 min, followed by a 2 min re-equilibration to the initial conditions. The injection volume was 10 μL. Electrospray ionization (ESI) was operated in positive mode (ESI+), and data were acquired in the sensitivity mode with a capillary voltage of 1 kV and a sampling cone voltage of 30 V. Leucine-enkephalin [M+H]^+^, *m*/*z* 556.2766, was used as the lock-mass compound for real-time mass correction. Data acquisition was performed in the data-independent acquisition (DIA) mode using high-definition MSE, which simultaneously collects data at a low collision energy (6 eV) and elevated collision energy (ramped from 20 to 40 eV). Argon was employed as the collision gas, while nitrogen served as the ion mobility separation (IMS) gas. The IMS gas flow rate was set at 25 mL/min, with a wave velocity of 250 m/s and an IMS pulse height of 45 V. Following the completion of the migration assays, the simulant samples were extracted and analyzed using UPLC-IMS-QTOF. Calibration curves for quantitation were prepared using standard solutions in ethanol. The structural identification of the detected peaks was achieved through UPLC-Q-TOF-MS analysis, utilizing accurate mass measurements. Molecular structures were determined using the ChemSpider chemical database (Royal Society of Chemistry 2025) by comparing the experimentally obtained accurate masses.

## 3. Results and Discussion

### 3.1. Identification of Volatile and Semi-Volatile Compounds by GC-MS

The optimization of PBAT and TPS/PBAT sample extraction was achieved through solvent extraction using ethanol and dichloromethane. This method was chosen because it detects a greater number of highly volatile compounds with a lower expansion volume [17,18,19,20]. Compound identification was analyzed via GC-MS using liquid injection. Figure 1 shows a chromatogram of the dissolved PBAT and TPS/PBAT films obtained through solvent extraction, facilitating the identification of unknown migrants in food contact materials. Table 2 lists 13 identified compounds with library matching scores ≥ 80% compared to NIST library. A total of 15 different volatile compounds were reported in the literature, with 7 found in PBAT resin and 9 in PBAT film. Both PBAT resin and film samples exhibited identical compounds when analyzed by GC–MS. Cyclopentanecarboxylic acid, 2-oxo-, ethyl ester is a degradation by-product of PBAT resins, as reported by Capolupo et al. [21] and Lin et al. [7]. Additionally, 1,6-Dioxacyclododecane-7,12-dione is a cyclic molecule formed during PBAT polymerization via polyester condensation involving the monomers 1,4-butanediol and adipic acid. It was found in both the resin and film and is classified as a non-intentionally added substance (NIAS) as noted by Lin et al. [7], Canellas et al. [22], Ubeda et al. [23]. Other compounds present at lower intensities in both PBAT resin and film included cyclopentanone, tetraethyl silicate, butylated hydroxytoluene (BHT), 2-ethylhexyl salicylate, methyl stearate, and ethyl stearate [19,24]. Cyclopentanone is formed by a radical reaction between aldehydes and ketones occurring in PBAT resin. The intensity of this peak slightly decreased in film samples, likely due to the high temperature during film manufacturing, as previously detected by Ubeda et al. [23] and Capolupo et al. [21]. Tetraethyl silicate, a silane compound, is a common additive used as a coupling agents in resins, adhesives, and inks to improve their chemical and thermal resistance [25]. BHT, used as an antioxidant or plastic stabilizer, is approved for food contact applications under EU regulation [26] and was previously detected by Ibarra et al. [6] and Pack et al. [27]. However, BHT has been scrutinized for its potential endocrine-disrupting properties and long-term safety at higher exposure levels [28]. 2-Ethylhexyl salicylate, used as a UV-filter additive, is generally regarded as safe but may contribute to cumulative exposure risks depending on its concentration and frequency of ingestion [29]. Stearate compounds were detected and are derived from fatty acids from, though they are also commonly found as contaminants in packaging materials. Additionally, an unknown compound with a retention time of 26.00 min was also detected. Based on the bibliographic information and mass spectrum several key fragments were observed: *m*/*z* 54.0 likely represents a small alkyl chain breakage associated with the molecule’s fragmentation, while *m*/*z* 104.0 could correspond to a larger fragment, indicating more complex molecular breakage possibly involving part of the cyclic structure. The mid-range *m*/*z* values of 132.0 and 149.0 may represent fragments of the dioxacyclododecane core, reflecting partial structures of the compound. Finally, *m*/*z* 369.1 represents the molecular ion peak (M+), which corresponds to the entire molecule and is characteristic of the molecular mass of 1,6-Dioxacyclododecane-7,12-dione. Given this fragmentation pattern, the compound is likely an oligomer formed during the PBAT polymerization process via polyester condensation, which involves the monomers 1,4-butanediol and adipic acid. This suggests that the compound is a larger oligomer than 1,6-Dioxacyclododecane-7,12-dione, which also has a similar fragmentation pattern. The presence of these specific *m*/*z* values indicates that the compound retains the core cyclic structure of 1,6-Dioxacyclododecane-7,12-dione but with additional polymerization units, resulting in a more complex molecular structure.

For TPS/PBAT films, both with and without food preservatives, the compound with the highest intensity identified was glycerol (CAS 56-81-5), which serves as an internal additive substance (IAS) and plasticizer for forming TPS via extrusion. Glycerol is also listed in the EU regulation No. 10/2011 [26] for food contact applications, either as an additive, a monomer, or for use in polymer production. Additionally, the results revealed five compounds not previously detected, including butyrolactone, 1,4-butanediol, 1,4-benzenedicarboxylic acid ethyl methyl ester, palmitic acid, and an unknown compound at a retention time of 9.10 min (fragments: *m*/*z* 61.0, 75.0, 108.0, 117.0, and 133.0). Palmitic acid is an additive used as a slip agent and it is also an approved additive and monomer for food contact material in the EU regulation [26]. The other compounds are likely degradation products formed during film manufacturing through reactions between ketones, aldehydes, and ethers. Compounds such as 1,6-dioxacyclododecane-7,12-dione and other cyclic esters have been previously reported in polyester-based materials and may lack comprehensive toxicological data [27,30,31,32]. Although substances such as glycerol and BHT are listed as approved additives under EU Regulation No. 10/2011, their migration levels must remain below specific limits to avoid potential endocrine-disrupting or cytotoxic effects [28,29]. The detection of palmitic acid and 2-ethylhexyl salicylate also warrants attention, as these may originate from additives or external contamination and contribute to cumulative dietary exposure. The incorporation of food preservatives to enhance the functionality of TPS/PBAT films also influenced the chemical structure of starch, PBAT, and TPS/PBAT matrix by acting as hydrolysis agents, plasticizers, and compatibilizers [3,12,33]. This structural modification may have promoted the breakdown of polymer chains and the formation of monomers and by-products.

### 3.2. Identification of Non-Volatile Compounds by UHPLC-Q-TOF-MS

The chromatograms of the TPS/PBAT film sample extraction, conducted as described in Section 2.3 and obtained using UHPLC-Q-TOF-MS, are presented in Figure 2. Structural elucidation of the detected peaks was performed using UPLC-Q-TOF-MS analysis by examining the mass spectra of each peak to determine their molecular composition. High-resolution mass measurements allowed for precise identification through the analysis of fragmentation patterns and isotopic distributions. The experimentally obtained masses were compared against entries in the ChemSpider chemical database to propose potential molecular structures [19]. Additionally, the mass spectra were cross-validated with previously reported data in the literature from Isella et al. [32], Aznar et al. [34] to confirm consistency with known compounds. Table 3 lists the non-volatile compounds identified in the total dissolution of PBAT resin, PBAT film, and TPS/PBAT with and without food preservatives, namely sodium nitrite (N), sodium erythorbate (E), sodium tripolyphisphate (T), sodium hexametaphosphate (H), and tetrasodium pyrophosphate (P). Most of the ten detected compounds, classified as NIAS, were cyclic compounds derived from adipic acid (AA), 1,4-butanediol (BD), and terephthalic acid (TPA) [19,22,34,35]. These are PBAT-derived oligomers, some of which were also identified via GC-MS, which likely originate as by-products from raw PBAT resin, formed during the polycondensation process of polyester synthesis. This is supported by the higher mass intensities observed in PBAT samples compared to TPS/PBAT, as shown in Figure 3. The intensities of these compounds decreased progressively throughout the manufacturing process, likely due to degradation caused by thermal processing during blown-film extrusion. PBAT film are subjected to a single extrusion step, whereas TPS/PBAT film undergo three heating stages—two during pellet preparation and one during film blowing—resulting in a more pronounced reduction in compound intensity [3,22,33,35]. Additionally, the incorporation of food preservatives introduced new compounds, including methyl phenylglyoxylate (4.69_165.0552), triacetin (5.13_203.0919), and resorcinol (6.08_111.0446), which function as curing agents, surface modifiers, and linking agents, respectively. These findings indicate that both thermal processing and the incorporation of preservatives significantly alter the chemical composition of TPS/PBAT films, potentially impacting their functionality and safety [36]. A thorough understanding of these changes is essential for optimizing film performance and ensuring their compliance with food packaging safety requirements, particularly in terms of chemical stability and migration behavior.

### 3.3. Migration and Risk Assessment

The migration of potential volatile compounds from active TPS/PBAT biodegradable packaging containing food preservatives into food simulants was investigated using headspace solid-phase microextraction in combination with a gas chromatography coupled to a mass spectrometric detection (HS-SPME–GC–MS). The food simulants used were simulant D2, simulant C, and simulant B, selected to predict the behavior of this packaging in contact with various types of food [19,36,38]. The analysis identified two volatile migrants after exposure to 95% ethanol: glycerin (10.33_92.05), 1,6 Dioxacyclododecane-7,12-dione (18.47_200.10), (Table 4). No migrants were detected in the 10% ethanol and 3% acetic acid simulants. The quantification of volatile compounds was performed using calibration curves generated from standard compounds. Quantification of 1,6-dioxacyclododecane-7,12-dione was performed using calibration curves derived from 1,4,7-trioxacyclotridecane-8,13-dione as the standard.

The limits of detection (LOD) and quantification (LOQ) are presented in Table 4. These values were determined using the signal-to-noise ratio (S/N) method, where the LOD corresponds to an S/N ratio of 3, and the LOQ corresponds to an S/N ratio of 10. The risk assessment was primarily based on the specific migration limits (SMLs) established by European Commission Regulation (EU) No 10/2011. For compounds not listed in the regulation’s positive list, the Threshold of Toxicological Concern (TTC) approach was applied. The TTC method is used to evaluate the safety of substances migrating from food contact materials when specific toxicological data are unavailable, as recommended by EFSA. Each compound was classified into one of the Cramer classes (I, II, or III) based on its chemical structure, with each class being assigned an exposure threshold [39]. The estimated exposures were compared to the TTC threshold and its derived SMLs. If the migration level remained below the derived SML, the compound was considered safe; otherwise, further toxicological evaluation would be required. In this study, the migrant 1,6 Dioxacyclododecane-7,12-dione was classified as Risk I (limit of Cramer Class I of 1.8 mg/person/day and TTC values for Class I), which generally indicates low toxicity. The TTC approach offers a conservative estimate of the acceptable daily intake by considering structural features and assigning a maximum intake level (TTC value) that is unlikely to pose health risks [25,26,34]. The migration level of 1,6-dioxacyclododecane-7,12-dione from active TPS/PBAT films was 9.14 ± 1.16 mg/kg, which did not exceed the TTC value under the assumed consumption scenario. Additionally, glycerin—listed in the positive list of Regulation (EU) No 10/2011—has a specific migration limit of 60 mg/kg. The observed migration levels of glycerin in all samples remained below this regulatory threshold.

Specific migration of non-volatile compounds from the active TPS/PBAT film was determined by UHPLC-Q-TOF-MS, as shown in Table 5. The analysis identified 33 non-volatile migrants, primarily consisting of linear and cyclic PBAT oligomers. A cyclic ester oligomer composed of diethylene glycol and adipic acid was used as a standard for oligomer quantification. Its purity and molecular structure were confirmed by nuclear magnetic resonance (NMR) analysis, conducted at the University of Zaragoza. As these oligomers are not listed in Commission Regulation (EU) No 10/2011 (2011), the Threshold of Toxicological Concern (TTC) approach was applied, as previously described. The results showed a maximum migration level of 0.98 ± 0.02 μg/g, which remained below the established safety threshold. Nevertheless, the toxicity of oligomers has become an issue of increasing concern. All of these compounds are classified as NIAS and their actual toxicological profiles remain largely unknown. Notably, the study also observed a reaction between the film and the high-fatty food simulant D2, indicating potential sensitivity to food polarity of the food matrix. However, the amount migrants generated remained below the regulatory limits.

To determine the migration behavior of TPS/PBAT films, a comparison with conventional petroleum-based plastics, such as polyethylene (PE), polypropylene (PP), and polyethylene terephthalate (PET), is essential. Studies have shown that conventional plastics also release various migrants, including both IAS, like antioxidants (e.g., Irganox 1010, BHT), and NIAS, such as cyclic oligomers and thermal degradation products [5,7,17,27,34,40]. For example, PET is known to leach cyclic oligomers (e.g., trimer and tetramer forms of ethylene terephthalate) and acetaldehyde under certain conditions, particularly in high-temperature or fatty food simulants [41,42]. In contrast, the TPS/PBAT films examined in this study exhibited a lower diversity of migratable substances, primarily consisting of known polymerization by-products and food-approved additives, such as glycerol, triacetin, and BHT. Furthermore, several identified oligomers in TPS/PBAT films were of biodegradable origin, and many were found in lower concentrations compared to those typically reported in PE- or PET-based materials [24,40,41,42]. These results suggest that TPS/PBAT films may offer a simplified and potentially less hazardous chemical profile, especially for applications involving food contact. However, further toxicological evaluations and long-term migration studies are needed to substantiate these preliminary advantages.

## 4. Conclusions

The developed TPS/PBAT films containing food preservatives (NaNO_2_, C_6_H_7_O_6_Na, Na_5_P_3_O_10_, Na_6_(PO_3_)_6_, and Na_4_P_2_O_7_), produced by blown film extrusion as active packaging, were evaluated for their potential application in food packaging and associated risk assessment. The potential risk posed by substances migrating from the developed TPS/PBAT active packaging was assessed using two analytical techniques: gas chromatography-mass spectrometry (GC-MS) and ultra-high-performance liquid chromatography-quadrupole time-of-flight mass spectrometry (UHPLC-Q-TOF-MS). These techniques enabled the identification of a broad spectrum of chemical compounds. The analysis identified 1,6-Dioxacyclododecane-7,12-dione, an oligomer from PBAT, as the major migrant. Additionally, various other compounds were detected, including potential additives, like plasticizers or antioxidants. The analysis also revealed the presence of a wide variety of non-intentionally added substances (NIAS), many of which likely resulted from degradation processes during extrusion and film formation. Migration experiments were conducted using three different food simulants to evaluate the potential migration behavior of these compounds. The experiments revealed the formation of new compounds formed through the interaction between food simulants and PBAT monomers or oligomers, particularly in the 95% ethanol simulant. These findings indicate that the active TPS/PBAT biodegradable film containing food preservatives have the potential to impact consumer safety and should be carefully evaluated for use in food contact materials.

## Figures and Tables

**Figure 1 foods-14-02171-f001:**
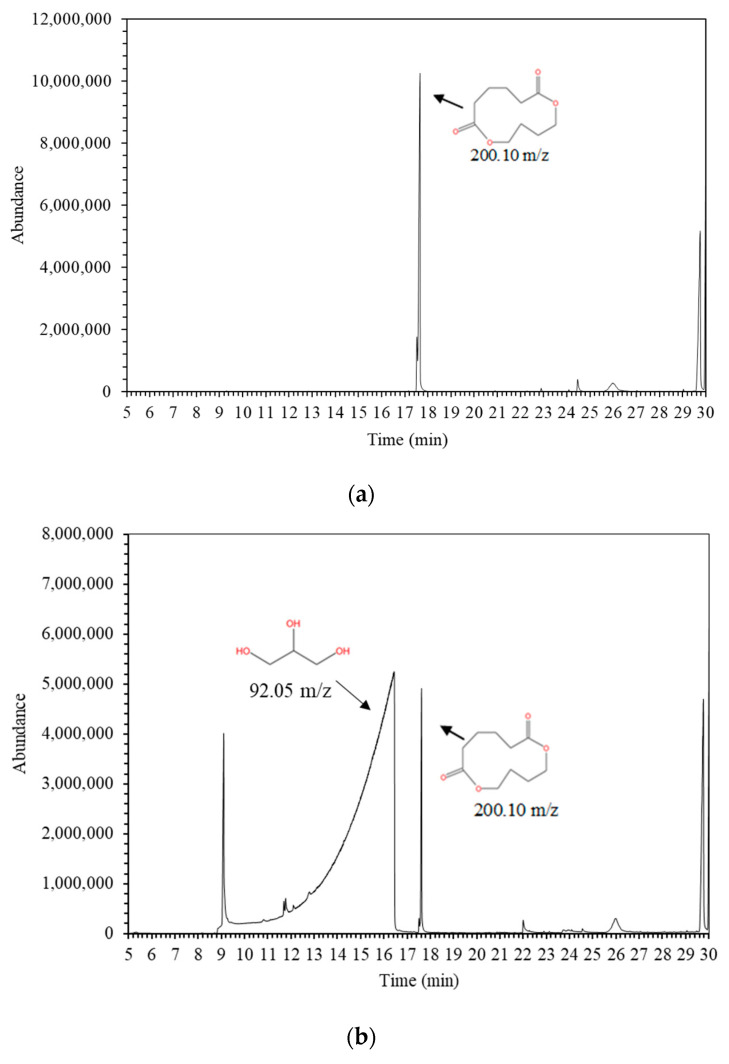
Chromatogram of (**a**) PBAT film and (**b**) TPS/PBAT blended film using GC-MS.

**Figure 2 foods-14-02171-f002:**
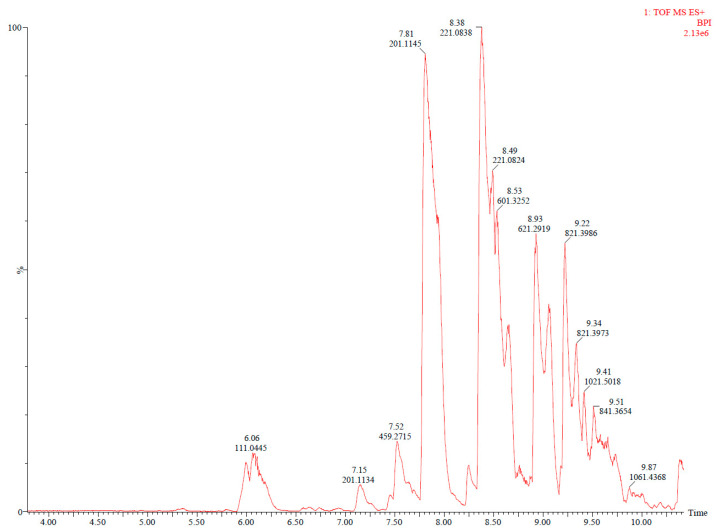
Chromatogram obtained by UHPLC-Q-TOF-MS analysis of TPS/PBAT film.

**Figure 3 foods-14-02171-f003:**
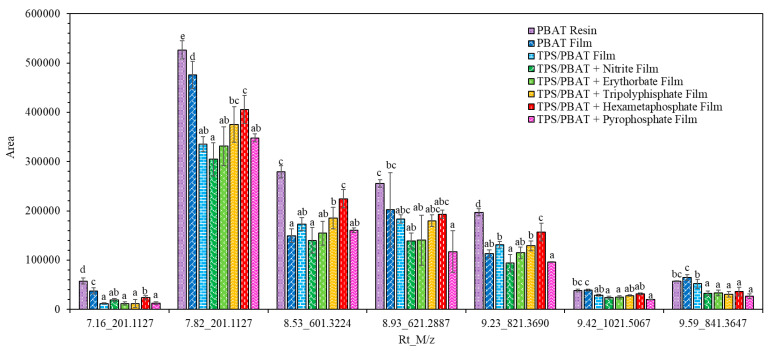
Intensity of the non-volatile components of PBAT resin, PBAT film, and TPS/PBAT with and without food preservatives, namely sodium nitrite, sodium erythorbate, sodium tripolyphisphate, sodium hexametaphosphate, and tetrasodium pyrophosphate films. Significant differences are indicated by different letters.

**Table 1 foods-14-02171-t001:** Name and characteristics of neat PBAT and PBAT/TPS film samples, which incorporated additives, namely sodium nitrite, sodium erythorbate, tetrasodium pyrophosphate, sodium tripolyphosphate and sodium hexametaphosphate, produced via blown-film extrusion.

Sample Name	Sample Code	Film Properties			
		Oxygen Permeability (cm^3^·mm/m^2^·day·atm)	Water Vapor Permeability (g·mm/m^2^·day·KPa)	Tensile Strength (MPa)	Elongation at Break (%)
PBAT Resins	R	-	-	-	-
PBAT Films	F	-	-	-	-
PBAT/TPS	C	16.69 ± 0.08	4.20 ± 0.14	14.43 ± 0.51	656.11 ± 11.50
PBAT/TPS + Sodium Nitrite	N	4.65 ± 0.26	4.51 ± 0.18	11.64 ± 0.66	794.62 ± 17.45
PBAT/TPS + Sodium Erythorbate	E	5.26 ± 0.18	3.54 ± 0.18	12.82 ± 0.58	651.94 ± 18.46
PBAT/TPS + Sodium Tripolyphosphate	T	21.42 ± 0.73	4.01 ± 0.06	10.60 ± 0.24	571.37 ± 16.18
PBAT/TPS + Sodium Hexametaphosphate	H	11.22 ± 1.89	3.11 ± 0.14	10.42 ± 8.78	595.95 ± 8.78
PBAT/TPS + Tetrasodium Pyrophosphate	P	29.07 ± 3.81	4.30 ± 0.20	9.31 ± 0.31	521.30 ± 6.19

**Table 2 foods-14-02171-t002:** Volatile and semi-volatile compounds were identified in total ethanol extracts of PBAT resin, PBAT film, and TPS/PBAT films—with and without incorporated food preservatives (sodium nitrite [N], sodium erythorbate [E], sodium tripolyphosphate [T], sodium hexametaphosphate [H], and tetrasodium pyrophosphate [P])—using GC–MS. The retention time (Rt) and relative intensity (I) of each compound were recorded, with intensity categorized by relative abundance as follows: I = 1 (0–5%), I = 2 (5–20%), I = 3 (20–50%), and I = 4 (50–100%).

No.	Rt	Candidate	CAS Number	Molecular Formular	Mass	Cramer Class	Samples								Remark
							R	F	C	N	E	T	H	P	
1	5.35	Cyclopentanone	120-92-3	C_5_H_8_O	84.06	II	1	1	1	1	1	1	nd	nd	Surfactant
2	7.73	Butyrolactone	96-48-0	C_4_H_6_O_2_	86.04	I	nd	nd	nd	1	1	nd	nd	nd	By product
3	8.86	1,4-Butanediol	110-63-4	C_4_H_10_O_2_	90.07	I	nd	nd	nd	1	1	1	nd	nd	Monomer
4	9.10	ni					nd	nd	1	1	1	1	1	1	Fragments; 61.0/75.0/108.0/117.0/133.0
5	9.28	Tetraethyl silicate	78-10-4	C_8_H_20_O_4_Si	208.11	III	1	1	nd	nd	nd	nd	nd	nd	Processing aids
6	12.85	Cyclopentanecarboxylic acid, 2-oxo-, ethyl ester	611-10-9	C_8_H_12_O_3_	156.18	-	nd	1	nd	1	nd	nd	nd	nd	By product
7	16.38	Glycerin	56-81-5	C_3_H_8_O_3_	92.05	I	nd	nd	4	4	4	4	4	4	Plasticizer
8	17.16	Butylated Hydroxytoluene	128-37-0	C_15_H_24_O	220.18	II	1	1	1	1	1	1	1	1	Antioxidants
9	17.55	1,6-Dioxacyclododecane-7,12-dione	777-95-7	C_10_H_16_O_4_	200.10	-	4	4	1	1	1	1	1	1	Oligomer
10	17.99	1,4-Benzenedicarboxylic acid, ethyl methyl ester	22163-52-6	C_11_H_12_O_4_	208.07	-	nd	nd	nd	1	1	1	nd	nd	By product
11	20.66	2-Ethylhexyl salicylate	118-60-5	C_15_H_22_O_3_	250.16	I	1	1	nd	nd	nd	1	1	nd	Processing aids
12	21.99	Palmitic acid	57-10-3	C_16_H_32_O_2_	256.24	I	nd	nd	1	nd	nd	1	1	1	FDA Inventory of Effective Food Contact Substance Notifications
13	23.50	Methyl stearate	112-61-8	C_19_H_38_O_2_	298.29	I	1	1	1	nd	1	nd	nd	nd	By product
14	24.09	Ethyl stearate	111-61-5	C_20_H_40_O_2_	312.30	I	1	1	1	nd	1	1	nd	nd	By product
15	26.00	ni					nd	1	1	1	1	1	1	1	Fragments; 54.0/104.0/132.0/149.0/369.1

nd: not detected, ni: not identified.

**Table 3 foods-14-02171-t003:** Non-volatile compounds identified in total dissolution of PBAT resin, PBAT film, and TPS/PBAT with and without food preservatives, namely sodium nitrite (N), sodium erythorbate (E), sodium tripolyphisphate (T), sodium hexametaphosphate (H), and tetrasodium pyrophosphate (H) films using UHPLC-Q-TOF-MS, where Rt is the retention time.

No	Rt	*m*/*z*	Adduct	Molecular Formula	Candidates	CAS No	Sample								Remark (Detected Similar by)
							R	F	C	N	E	T	H	P	
1	4.69	165.0552	1H+	C_9_H_9_O_3_	Methyl phenylglyoxylate	15206-55-0					x				Lin et al. (2023) [7]
2	5.13	203.0919	1H+	C_9_H_15_O_5_	Triacetin	102-76-1				x	x	x	x	x	Lin et al. (2023) [7]
3	6.08	111.0446	1H+	C_6_H_6_O_2_	Resorcinol	108-46-3	x	x	x	x	x	x	x	x	Lin et al. (2023) [7]
4	7.16	201.1127	1H+	C_10_H_16_O_4_	Cyclic [AA-BD]	777-95-7	x	x	x	x	x	x	x	x	Canellas, Vera & Nerín (2015) [22] and Aznar, Ubeda, Dreolin & Nerín, 2019 [34]
5	7.82	201.1127	1H+	C_10_H_16_O_4_	Cyclic [AA-BD]	777-95-7	x	x	x	x	x	x	x	x	Canellas, Vera & Nerín (2015) [22] and Aznar, Ubeda, Dreolin & Nerín, 2019 [34]
6	8.53	601.3224	1H+	C_30_H_48_O_12_	Cyclic [AA-BD]_3_		x	x	x	x	x	x	x	x	Zhang, Su, Shang, Weng & Zhu (2023) [35]
7	8.93	621.2887	1H+	C_32_H_44_O_12_	Cyclic [TPA-AA_2_-BD_2_]		x	x	x	x	x	x	x	x	Osorio, Aznar, Nerín, Elliott & Chevallier (2022) [37] and Zhang, Su, Shang, Weng & Zhu (2023) [35]
8	9.23	821.396	1H+	C_42_H_60_O_16_	Cyclic [TPA-AA_3_-BD_3_]		x	x	x	x	x	x	x	x	Osorio, Aznar, Nerín, Elliott & Chevallier (2022) [37] and Zhang, Su, Shang, Weng & Zhu (2023) [35]
9	9.42	1021.5067	1H+	C_52_H_76_O_20_	Cyclic [TPA-AA_4_-BD_4_]		x	x	x	x	x	x	x	x	Osorio, Aznar, Nerín, Elliott & Chevallier (2022), [37] and Zhang, Su, Shang, Weng & Zhu (2023) [35]
10	9.59	841.3647	1H+	C_44_H_56_O_16_	Cyclic [TPA_2_-AA_2_-BD_4_]		x	x	x	x	x	x	x	x	Zhang, Su, Shang, Weng & Zhu (2023) [35]

x: detected.

**Table 4 foods-14-02171-t004:** Specific volatile and semi-volatile migration from active TPS/PBAT with and without food preservatives, namely sodium nitrite (N), sodium erythorbate (E), sodium tripolyphisphate (T), sodium hexametaphosphate (H), and tetrasodium pyrophosphate (P) films to ethanol 95% (V/V), expressed as mg of analyte per kg of food simulant (±standard deviation).

No	Rt	Migrants	CAS No.	Molecular Formular	Mass	Cramer Class	LOD (μg/g)	LOQ (μg/g)	Specific Migration (mg/kg)					
									TPS/PBAT	N	E	T	H	P
1	10.33	Glycerin	56-81-5	C_3_H_8_O_3_	92.05	I	0.20	0.70	1.20	1.30	1.20	1.10	1.10	1.40
2	18.47	1,6-Dioxacyclododecane-7,12-dione	777-95-7	C_10_H_16_O_4_	200.10	I	1.16	3.51	9.08 ± 0.77	9.14 ± 1.16	4.07 ± 0.06	<3.51	<3.51	<3.51

**Table 5 foods-14-02171-t005:** Specific non-volatile migration from active TPS/PBAT with and without food preservatives, namely sodium nitrite (N), sodium erythorbate (E), sodium tripolyphisphate (T), sodium hexametaphosphate (H), and tetrasodium pyrophosphate (H) films to ethanol 95% (*V*/*V*), expressed as μg of analyte per g of food simulant (±standard deviation).

No.	Formular	Compound Name	Retention Time (min)	CAS	Adduct	*m*/*z*	Cramer Class	LOD (μg g^−1^)	LOQ (μg g^−1^)	Specific Migration in EtOH (μg g^−1^)					
										TPS/PBAT	N	E	T	H	P
1	C_10_H_16_O_4_	Cyclic [AA-BD]	6.19	777-95-7	1H^+^	201.1126	I	0.08	0.23	<LOQ	<LOQ	0.23 ± 0.01	<LOQ	<LOQ	<LOQ
2	C_14_H_26_O_6_	Linear [AA-BD_2_]	6.25	20985-13-1	23Na^+^	313.1623	I	0.08	0.23	<LOQ	<LOQ	1.04 ± 0.04	<LOQ	0.91 ± 0.04	0.26 ± 0.02
3	C_12_H_22_O_5_	CH_3_CH_2_OH-Linear [AA-BD]	6.77	925-06-4	23Na^+^	269.1365	I	0.08	0.23	<LOQ	0.24 ±				
4	C_10_H_18_O_4_	Sebacic acid	7.37	141-28-6	23Na^+^	225.1103	I	0.11	0.34		<LOQ		<LOQ		<LOQ
5	C_24_H_42_O_10_	Linear [BD_3_-AA_2_]	7.56		23Na^+^	513.2667	I	0.08	0.23		0.29 ± 0.01		0.25 ± 0.04		0.26 ± 0.01
6	C_10_H_10_O_4_	CH_3_CH_2_OH-TPA	7.68	131-11-3	1H^+^	195.065	I	0.17	0.50						
7	C_22_H_30_O_9_	Linear [TPA-AA-BD_2_]	7.81		23Na^+^	461.1788	I	0.17	0.50		<LOD				
8	C_20_H_32_O_8_	Cyclic [AA_2_-BD_2_]	7.93	78837-87-3	23Na^+^	423.1995	I	0.08	0.23	0.88 ± 0.02	0.98 ± 0.02	0.94 ± 0.04	0.87 ± 0.04	0.84 ± 0.02	0.84 ± 0.01
9	C_22_H_38_O_9_	CH_3_CH_2_OH-Linear [AA_2_-BD_2_]	7.98		23Na^+^	469.2414	I	0.08	0.23						
10	C_26_H_38_O_10_	Linear [TPA-AA-BD_3_]	8.05		23Na^+^	533.2363	I	0.17	0.50	<LOQ	<LOD	<LOD	<LOD		<LOD
11	C_34_H_58_O_14_	Linear [AA_3_-BD_4_]	8.22		23Na^+^	713.3724	I	0.08	0.23		<LOQ		<LOQ		<LOQ
12	C_24_H_34_O_9_	CH_3_CH_2_OH-Linear [TPA-AA-BD_2_]	8.45	21259–20–1	23Na^+^	489.2101	I	0.17	0.50						
13	C_22_H_28_O_8_	Cyclic [TPA-BD_2_-AA]	8.49		1H^+^	421.1862	I	0.17	0.50	<LOQ	<LOD	<LOD	<LOD	<LOD	<LOD
14	C_20_H_26_O_8_	CH_3_CH_2_OH-linear [AA-BB-TPA]	8.54		1H^+^	395.1706	I	0.17	0.50						
15	C_30_H_48_O_12_	Cyclic [AA_3_-BD_3_]	8.63	1135871-5-6	23Na^+^	623.3043	I	0.08	0.23	0.26 ± 0.02	0.33 ± 0.02	0.35 ± 0.02	0.28 ± 0.02	0.32 ± 0.02	0.27 ± 0.00
16	C_24_H_24_O_8_	Cyclic [TPA-BD_2_]	8.86		23Na+	441.1558	I	0.17	0.50			<LOQ			
17	C_32_H_44_O_12_	Cyclic [TPA-BD_3_-AA_2_]	9.02		23Na^+^	643.273	I	0.17	0.50	<LOQ	<LOD	<LOD	<LOD	<LOD	<LOD
18	C_44_H_66_O_17_	Cyclic [TPA-BD_3_-AA_4_]	9.19		23Na^+^	889.4198	I	0.17	0.50	<LOQ	<LOD	<LOD	<LOD	<LOD	<LOD
19	C_42_H_60_O_16_	Cyclic [TPA-BD_4_-AA_3_]	9.30		23Na^+^	843.3779	I	0.17	0.50	<LOQ	<LOD	<LOD	<LOD	<LOD	<LOD
20	C_22_H_28_O_8_	Cyclic [TPA-BD_3_-AA_2_]	9.42		1H^+^	443.1688	I	0.17	0.50		<LOD				<LOD
21	C_34_H_40_O_12_	Cyclic [TPA_2_-BD_3_-AA_1_]	9.44		23Na^+^	663.2417	I	0.17	0.50	<LOQ	<LOD	<LOD		<LOD	
22	C_44_H_56_O_16_	Cyclic [TPA_2_-BD_4_-AA_2_]	9.57		23Na^+^	863.3466	I	0.17	0.50			<LOD		<LOD	
										**Specific Migration in 10% EtOH (μg g^−1^)**					
23	C_10_H_18_O_5_	Linear [AA-BD]	5.42	777-95-7	23Na^+^	241.1036	I	0.08	0.23	<LOQ	<LOQ		<LOQ		
24	C_14_H_26_O_6_	Linear [AA-BD_2_]	6.25	20985-13-1	23Na^+^	313.1628	I	0.08	0.23	<LOQ	<LOQ				
25	C_14_H_17_O_7_	CH3CH2OH-Linear TPA-BD	6.70		1H+	299.1102	I	0.17	0.50						<LOQ
26	C_16_H_26_O_8_	Linear [AA_2_-BD]	6.83		23Na^+^	396.1526	I	0.08	0.23				<LOQ		
27	C_20_H_34_O_9_	Linear [AA_2-_BD_2_]	7.24		23Na^+^	441.2095	I	0.08	0.23	<LOQ	0.25 ± 0.08	<LOQ		<LOQ	
28	C_12_H_120_O_4_	Cyclic [TPA-BD]	7.48		H+	221.0809	I	0.17	0.50	<LOQ	<LOQ		<LOQ	<LOQ	
29	C_24_H_42_O_10_	Linear [AA_2-_BD_3_]	7.54		23Na^+^	513.2678	I	0.08	0.23		<LOQ				
30	C_20_H_32_O_8_	Cyclic [AA_2_-BD_2_]	7.91	78837-87-3	23Na^+^	423.1971	I	0.08	0.23	0.35 ± 0.11	0.37 ± 0.10	0.57 ± 0.08		0.31 ± 0.08	
										**Specific Migration in 3% HAc (μg g^−1^)**					
31	C_6_H_6_O_2_	Resorcinol	3.91	108-46-3	1H^+^	111.0435	I	-	-	x	x	x	x	x	-
32	C_10_H_18_O_5_	Linear [AA-BD]	5.42	777-95-7	23Na^+^	241.1030	I	0.08	0.23	<LOQ	<LOQ	<LOQ	<LOQ	<LOQ	<LOQ
33	C_16_H_28_O_7_	HAc-Linear [AA-BD]	7.00		1H^+^	333.1317	I	0.08	0.23	<LOQ	<LOQ	<LOQ	<LOQ	<LOQ	<LOQ

## Data Availability

The original contributions presented in the study are included in the article, further inquiries can be directed to the corresponding author.

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
