# Peer review of "Screening and Relative Quantification of Migration from Novel Thermoplastic Starch and PBAT Blend Packaging"

_foods, 2025, doi:10.3390/foods14132171_

Round 1

Reviewer 1 Report

Comments and Suggestions for Authors

The article comprehensively addresses the analysis and risk assessment of substances migrating from biodegradable TPS/PBAT films into food simulants, as used in food packaging applications. The study provides significant data for evaluating the safety and sustainability of the innovative TPS/PBAT blend material. However, there are some methodological and presentation-related shortcomings that need to be addressed:

  • Additional information should be provided regarding the optimization of extraction and migration protocols.

  • The potential health risks of newly identified compounds, particularly those detected in 95% ethanol simulant, should be discussed with reference to the relevant literature.

  • A comparative analysis of migration profiles with conventional plastics would be beneficial to highlight the advantages of the material.

  • The main findings should be presented more prominently in the abstract (e.g., by stating that "no regulatory migration limits were exceeded").

  • Comparing the migration profiles of TPS/PBAT films with those of conventional plastics (e.g., LDPE) would further strengthen the innovative aspect of the study.

  • The choice of extraction temperature (80°C) and duration (30 minutes) for the SPME fiber (DVB/CAR/PDMS) should be justified with reference to the literature.

  • The manuscript makes an important contribution to the application of biodegradable TPS/PBAT films in food packaging. However, it could be significantly strengthened by clarifying methodological details and providing a more comprehensive discussion of the results. With the suggested revisions, the study will offer greater scientific impact and practical relevance.

Author Response

Reviewer 1:

The article comprehensively addresses the analysis and risk assessment of substances migrating from biodegradable TPS/PBAT films into food simulants, as used in food packaging applications. The study provides significant data for evaluating the safety and sustainability of the innovative TPS/PBAT blend material. However, there are some methodological and presentation-related shortcomings that need to be addressed:

  • Additional information should be provided regarding the optimization of extraction and migration protocols.

Additional details have been included to clarify the optimization of the extraction and migration protocols.

  • The potential health risks of newly identified compounds, particularly those detected in 95% ethanol simulant, should be discussed with reference to the relevant literature.

A discussion on the potential health risks of the newly identified compounds has been added to the manuscript with appropriate references to the relevant literature

  • A comparative analysis of migration profiles with conventional plastics would be beneficial to highlight the advantages of the material.

The manuscript has been revised to include a comparative analysis of migration profiles with conventional plastics, highlighting the advantages of the material

  • The main findings should be presented more prominently in the abstract (e.g., by stating that "no regulatory migration limits were exceeded").

Abstract has been revised: Novel biodegradable food packaging material based on cassava thermoplastic starch (TPS) and polybutylene adipate terephthalate (PBAT) blends containing food preservatives via blown-film extrusion was successfully developed to improve functionality to enhance appearance, taste, color and delay quality deterioration of the food products. However, the addition of food preservatives directly affects consumer perception, health and safety. This research aim to assess the risk of both intentionally and non-intentionally added substances occurring in developed active packaging. The migration of both intentionally and non-intentionally added substances was then evaluated using GC-MS and UHPLC-Q-TOF-MS. Fifteen different volatile compounds were detected, the main compound was 1,6-Dioxacyclododecane-7,12-dione, originating from PBAT. The polymerization of adipic acid, terephthalic acid, and butanediol resulted in the formation of linear and cyclic oligomers of PBAT. Migration experiments were conducted with three food simulants (95% ethanol, 10% ethanol, and 3% acetic acid) for 10 days at 60 °C. No migration above the detection limit of our analytical methods was detected in the case of 3% acetic acid and 10% ethanol. However, migration studies with 95% ethanol revealed new compounds formed from the reaction between food simulants and PBAT monomers and oligomers, indicating sensitivity to high-polarity food simulants. Nevertheless, the amount of these compounds generated did not exceed the regulatory migration limits.”

  • Comparing the migration profiles of TPS/PBAT films with those of conventional plastics (e.g., LDPE) would further strengthen the innovative aspect of the study.

The manuscript has been revised and compared with conventional plastics.

  • The choice of extraction temperature (80°C) and duration (30 minutes) for the SPME fiber (DVB/CAR/PDMS) should be justified with reference to the literature.

Preliminary extractions were per-formed at 80 °C for 30 minutes to enhance analyte partitioning into the headspace without compromising thermal stability. This information has been added to the manuscript.

  • The manuscript makes an important contribution to the application of biodegradable TPS/PBAT films in food packaging. However, it could be significantly strengthened by clarifying methodological details and providing a more comprehensive discussion of the results. With the suggested revisions, the study will offer greater scientific impact and practical relevance.

Thank you for your valuable feedback and for taking the time to review. The manuscript has been revised following your suggestions.

Reviewer 2 Report

Comments and Suggestions for Authors

Comments.
1. The abstract lacks critical context (background) and fails to highlight the novelty of the work explicitly. Please provide the background and the novelty of this work. 
2. On page 4, the authors stated that Cyclopentanecarboxylic acid, 2-oxo-, ethyl ester is a degradation by-product of PBAT resins, as reported by Capolupo, et al. [13] and Lin, Wu, Zhong, Xian, Zhong, Dong, Liang, Hu, Wu, Yang, Sui and Zhou [7]. Please check the citation formation of [7].
3. In Figure 1, the GC-MS peaks should be labeled with m/z.
4. In Figure 3, the difference between PBAT-based materials should be considered (P<0.05).
5. The formation of Table 4 should be reorganised with a good presentation.
6. For evaluating the safety of different films, the authors should provide the results of Biocompatibility testing.

Author Response

  1. The abstract lacks critical context (background) and fails to highlight the novelty of the work explicitly. Please provide the background and the novelty of this work. 

Abstract has been revised as Novel biodegradable food packaging material based on cassava thermoplastic starch (TPS) and polybutylene adipate terephthalate (PBAT) blends containing food preservatives via blown-film extrusion was successfully developed to improve functionality to enhance appearance, taste, color and delay quality deterioration of the food products. However, the addition of food preservatives directly affects consumer perception, health and safety. This research aim to assess the risk of both intentionally and non-intentionally added substances occurring in developed active packaging.  The migration of both intentionally and non-intentionally added substances was then evaluated using GC-MS and UHPLC-Q-TOF-MS. Fifteen different volatile compounds were detected, the main compound was 1,6-Dioxacyclododecane-7,12-dione, originating from PBAT. The polymerization of adipic acid, terephthalic acid, and butanediol resulted in the formation of linear and cyclic oligomers of PBAT. Migration experiments were conducted with three food simulants (95% ethanol, 10% ethanol, and 3% acetic acid) for 10 days at 60 °C. No migration above the detection limit of our analytical methods was detected in the case of 3% acetic acid and 10% ethanol. However, migration studies with 95% ethanol revealed new compounds formed from the reaction between food simulants and PBAT monomers and oligomers, indicating sensitivity to high-polarity food simulants. Nevertheless, the amount of these compounds generated did not exceed the regulatory migration limits.”

  1. On page 4, the authors stated that Cyclopentanecarboxylic acid, 2-oxo-, ethyl ester is a degradation by-product of PBAT resins, as reported by Capolupo, et al. [13] and Lin, Wu, Zhong, Xian, Zhong, Dong, Liang, Hu, Wu, Yang, Sui and Zhou [7]. Please check the citation formation of [7].

The citations for Capolupo et al. [13] and Lin et al. [7] have been revised, and all references have been checked.

  1. In Figure 1, the GC-MS peaks should be labeled with m/z.

Figure 1 has been updated to include m/z labels on the GC-MS peaks

  1. In Figure 3, the difference between PBAT-based materials should be considered (P<0.05).

Figure 3 has been updated to include statistical analysis (P < 0.05) to highlight the differences between PBAT-based materials.

  1. The formation of Table 4 should be reorganized with a good presentation.

Table 4 has been reorganized.

  1. For evaluating the safety of different films, the authors should provide the results of Biocompatibility testing.

Thank you for your valuable comment. We agree that biocompatibility testing is essential for evaluating the safety of the developed films, particularly for applications involving direct contact with food or biological systems. Although our study focused on the physicochemical and functional properties of the films, biocompatibility testing was not conducted. We acknowledge this limitation and will address it in future work to further validate the films' safety and applicability.

Round 2

Reviewer 2 Report

Comments and Suggestions for Authors

Comments
The manuscript is ok, but the "Table 5" should be changed to "Table 5".